# Trend in Hypertension Prevalence and Health Behaviors among the Brazilian Adult Population: 2006–2019

**Thaís C. M. Caldeira** [1,*] ![ID], **Ana Claudia R. A. Sereno** [2], **Marcela M. Soares** [1], **Emanuella G. Maia** [3] **and Rafael M. Claro** [2] ![ID]

1    Department of Preventive and Social Medicine, Universidade Federal de Minas Gerais, Belo Horizonte 30130-100, Brazil
2    Nutrition Department, Universidade Federal de Minas Gerais, Belo Horizonte 30130-100, Brazil; rafael.claro@gmail.com (R.M.C.)
3    Health Sciences Department, State University of Santa Cruz, Ilhéus 45662-900, Brazil
*    Correspondence: thaismarquezinec@gmail.com; Tel.: +55-319-9129-8297

**Abstract:** Our objective was to analyze temporal trends in the prevalence of self-reported hypertension among Brazilian adults and to investigate differences in health behaviors between individuals with and without hypertension between 2006 and 2019. Data from the Surveillance System for Risk Factors and Protection for Chronic Diseases by Telephone Survey were analyzed (n = 730,309). Prais–Winsten regression was used to identify linear trends in the prevalence of hypertension for the entire period (2006–2019) and for the past 5 years. Poisson regression models were used to investigate the differences in health behaviors among individuals with and without hypertension. The prevalence of hypertension (approximately 24.0%) remained stable from 2006 to 2019 and decreased from 25.1% to 24.6% from 2015 to 2019. In the adjusted analyses, individuals with hypertension showed a significant association with unhealthy lifestyle habits: lower recommended intake of fruits and vegetables (APR = 0.97; *p* = 0.022), lower regular intake of fruits (APR = 0.98; *p* < 0.001), lower regular intake of beans (APR = 0.97; *p* < 0.001), lower leisure-time exercising (APR = 0.89; *p* < 0.001), higher abusive consumption of alcoholic beverages (APR = 1.04; *p* = 0.004), higher prevalence of overweight (APR = 1.40; *p* < 0.001), and higher prevalence of obesity (APR = 2.17; *p* < 0.001). Hypertension prevalence has remained stable during the entire period and decreased in the most recent period. Individuals with hypertension reported unfavorable scenarios for healthy habits.

**Keywords:** hypertension; lifestyles; public health; multivariate analysis





## 1. Introduction

Hypertension is the most prevalent preventable risk factor for cardiovascular diseases (CVD), which are the leading cause of death in the Region of the Americas, with 2.0 million deaths from CVC in 2019 [1]. Hypertension is a multifactorial condition featured by high and sustained blood pressure levels [2], which, when associated with other risk factors, such as behavioral (unhealthy diet, physical inactivity, smoking, and alcohol abuse) and metabolic ones (overweight or obesity and diabetes), leads to worsened prognosis in individuals with CVD [2–4]. Approximately 32% of the global adult population was affected by hypertension in 2010 [5], and this prevalence has increased mainly among low- and middle-income countries over the past decades [6]. In Brazil, this condition affected 24% of adult individuals [7], and it was responsible for approximately 16% of total deaths from CVD in 2019 [8].

Controlling and reducing the prevalence of hypertension by 25% in the population between 2015 and 2025 is one of the goals of the "Global Action Plan for the Prevention and Control of NCDs 2013–2020" by the World Health Organization (WHO) [9]. Several health promotion measures focused on hypertension control were suggested to achieve

this goal. Among them, one finds decreasing sodium content in industrialized food products, encouraging the practice of exercising and free distribution of hypertension control drugs [9]. The "Strategic Action Plan to Tackle Noncommunicable Diseases (NCD) in Brazil 2011–2022", by the Ministry of Health (MoH) does not provide specific targets regarding the prevalence of hypertension in the country, but it presents guidelines to manage and monitor hypertension, comorbidities, and their determining factors, in addition, to set population targets for behaviors capable of benefiting people with hypertension, such as physical activity, tobacco reduction and healthy eating [10]. Even with the update of the plan (for 2021 to 2030), more concrete goals were not planned [11].

Monitoring the prevalence of hypertension, and its risk factors, can contribute to the development of more effective health policies [5,10]. Thus, information for the continuous monitoring of the frequency and distribution of risk and protective factors for NCD and self-reported morbidity has been collected through a telephone survey conducted by MoH since 2006 [12]. However, only a small amount of data has been analyzed so far, and it resulted in an incomplete set of information. The analysis of these data allows us to present the Brazilian scenario regarding health actions for the control of Hypertension. Given this scenario, the objective of the current study was to analyze the temporal trends of self-reported hypertension prevalence by Brazilian adults over a 14-year period of time and to investigate the differences in health behaviors between individuals with and without hypertension.

## 2. Materials and Methods

Time series analysis and cross-sectional study were conducted with data collected by the Surveillance System for Risk and Protection Factors for Chronic Diseases by Telephone Survey (Vigitel) from 2006 to 2019. Vigitel is an annual survey based on interviews conducted through landline phone calls to adult individuals ($\geq$18 years old) living in the 26 Brazilian state capitals and the Federal District [12].

Vigitel's sampling process is divided into two stages. The first stage comprises the random selection of five thousand landlines in each city from the electronic register of residential landlines of the main telephone companies in the country. These numbers are organized into replicas with 200 landlines (it is performed by reproducing the same proportion of lines as the original registration). Once the landline eligibility is established, one adult individual among the residents of each household is selected (simple random sample) and invited to participate in the survey (second stage). The minimum sample size of approximately 2000 individuals to be interviewed each year in each city was established to estimate the frequency of each indicator at a 95% confidence interval (95%CI) and a maximum error of two percentage points. Samples of approximately 1500 individuals are accepted for cities with low landline coverage [12].

All Vigitel estimates are weighted to represent the total adult population of each city. The final weight comprises two factors; one of them deals with the likelihood of unequal sampling of households with more than one landline and more than one resident, whereas the second factor compares the distribution of the interviewed population to that projected for the entire population in each study site and year (according to sex, age, and schooling, based on the Rake method), based on official projections for the population [12]. Further details about the sampling process used in Vigitel can be found in the system's annual reports [12].

A computer-assisted telephone interviewing method was used, allowing the immediate identification of invalid responses and the automatic pass-through of not-applicable questions ensuring the continuous feeding of the database. Questions concerning the self-reported medical diagnosis of hypertension and health behaviors linked to NCD were used in the present investigation. The presence of hypertension was established based on an affirmative response to the question, "Has a doctor ever told you that you have high pressure?" (Yes | No). The health behaviors were evaluated through questions about food consumption, physical activity, alcohol intake, smoking, overweight and obesity,

and mean of (kg/m$^2$) Body Mass Index (BMI). Among the fourteen indicators, six of them were classified as protective factors: (i) Recommended intake of fruits and vegetables ($\geq$5 portions/day on $\geq$5 days/week); (ii) Regular intake of fruits and vegetables ($\geq$5 times/day); (iii) Regular intake of fruits ($\geq$5 days/week); (iv) Regular intake of vegetables ($\geq$5 days/week); (v) Regular intake of beans ($\geq$5 days/week); and (vi) Leisure-time exercising ($\geq$150 min/week (moderate intensity) or 75 min/week (vigorous intensity)) [13]. Eight behaviors were classified as risk factors: (i) Fat-rich meat intake (fat-rich red meat or poultry skin on $\geq$1 day/week); (ii) Whole milk intake ($\geq$1 day/week); (iii) Regular intake of soft drinks and sugar-sweetened beverages ($\geq$5 days/week); (iv) Regular intake of sweets ($\geq$5 days/week); (v) Abusive consumption of alcoholic beverages ($\geq$4 drinks for women or $\geq$5 drinks for men in a single day); (vi) Smoking (Yes, regardless of frequency and quantity); (vii) Overweight (BMI $\geq$ 25 kg/m$^2$); and (viii) Obesity (BMI $\geq$ 30 kg/m$^2$).

Not all indicators were made available in all years due to changes in the Vigitel questionnaire during the investigated period. The indicators and the period when each indicator was available and their respective questions were presented in the Supplementary Materials.

A set of sociodemographic characteristics of individuals complemented the analysis, such as sex (male and female), age group (18–24, 25–34, 35–44, 45–54, 55–64, and $\geq$ 65 years old) and number of school years (0–8, 9–11 and $\geq$ 12 years).

*Statistical Analysis*

The investigated population was initially described for each of the analyzed years based on their sociodemographic characteristics (sex, age group, and schooling). Then, hypertension prevalence was estimated for each year—smoothed by a moving average—for the full set of population and sociodemographic characteristics. The use of moving average requires creating a new series whose values comprise the average of gross observations in the original time series (allowing data smoothing and reducing likely noises between collections). An unweighted time window of the 1:1:1 (three years) type was selected in the current study. Thus, the original value recorded for hypertension rate in the population in 2008 was replaced by the arithmetic mean of hypertension rates recorded in 2007, 2008, and 2009, for example. Prais–Winsten linear regression models were used to identify significant trends to increase or decrease hypertension prevalence in the analyzed period. This regression method was adopted because it takes into consideration the incidence of serial autocorrelation [14]. Coefficients resulting from these models have indicated changes in percentage points per year (pp/year) of the indicator in the analyzed period. Significant values observed for this coefficient (*p*-value < 0.05) have indicated consistent and significant variations in the time series. This analysis was applied to the total investigated period (2006–2019) and to the most recent period (2015–2019).

Subsequently, the frequency of health behaviors presented by individuals with and without hypertension was identified for the whole population and stratified by sex. Poisson regression models were used to calculate gross and adjusted prevalence ratios (based on sociodemographic characteristics). The health behaviors were analyzed as dependent variables, whereas the prevalence of hypertension was used as independent variable. We considered the prevalence ratio significant when *p*-value was less than 0.05.

Data were organized and analyzed in the Stata software 16.1 (Stata Corporation; 2019. Vigitel's databases are available for public use at the official MoH website Available online: http://svs.aids.gov.br/download/Vigitel/ (accessed on 10 January 2023).

## 3. Results

A total of 730,309 adults were interviewed between 2006 and 2019. The prevalence of adults in the age group 18–24 years decreased by 0.38 percentage points per year (pp/year) from 2006 to 2019; it ranged from 18.9% to 13.8%. On the other hand, the prevalence of adults in the age group 45 years or older increased from 35.2% to 42.4% in the same period. There was a decrease of 1.32 pp/year in the prevalence of adults with 0 to 8 years of school

(from 45.5% to 28.8%), as well as an increase of 1.01 pp/year among those with 12 or more years of school (from 21.2% to 32.8%). Trends regarding age and schooling remained unchanged in the most recent period—from 2015 to 2019 (Table 1).

**Table 1.** Distribution of the adult population ($\geq$18 years old) in the capitals of the 26 Brazilian states and the Federal District, according to sociodemographic characteristics. Vigitel Brazil, 2006–2019.

| Variables | 2006 | 2007 | 2008 | 2009 | 2010 | 2011 | 2012 | 2013 | 2014 | 2015 | 2016 | 2017 | 2018 | 2019 | Coef. [†] (2006/19) | Coef. [†] (2015/19) |
|---|---|---|---|---|---|---|---|---|---|---|---|---|---|---|---|---|
| **Sex** | | | | | | | | | | | | | | | | |
| Male | 46.1 | 46.2 | 46.1 | 46.1 | 46.1 | 46.1 | 46.1 | 46.1 | 46.1 | 46.0 | 46.0 | 46.0 | 46.0 | 46.0 | −0.01 ** | −0.02 ** |
| Female | 53.9 | 53.8 | 53.9 | 53.9 | 53.9 | 53.9 | 53.9 | 53.9 | 53.9 | 54.0 | 54.0 | 54.0 | 54.0 | 54.0 | 0.01 ** | 0.02 ** |
| **Age group, years** | | | | | | | | | | | | | | | | |
| 18 to 24 | 18.9 | 18.2 | 17.9 | 17.5 | 17.1 | 16.7 | 16.4 | 15.9 | 15.6 | 15.2 | 14.8 | 14.5 | 14.1 | 13.8 | −0.38 ** | −0.36 ** |
| 25 to 34 | 25.4 | 25.4 | 25.4 | 25.4 | 25.4 | 25.4 | 25.2 | 25.3 | 25.3 | 25.2 | 25.2 | 25.2 | 25.1 | 25.0 | −0.03 ** | −0.05 ** |
| 35 to 44 | 20.6 | 20.5 | 20.4 | 20.2 | 20.1 | 20.0 | 19.9 | 19.7 | 19.6 | 19.4 | 19.3 | 19.1 | 19.0 | 18.8 | −0.14 ** | −0.15 ** |
| 45 to 54 | 15.8 | 15.9 | 16.1 | 16.3 | 16.4 | 16.6 | 16.6 | 16.9 | 16.9 | 17.1 | 17.3 | 17.6 | 17.7 | 17.9 | 0.16 ** | 0.16 ** |
| 55 to 64 | 10.0 | 10.2 | 10.4 | 10.7 | 10.9 | 11.1 | 11.4 | 11.6 | 11.8 | 12.1 | 12.3 | 12.6 | 12.8 | 13.1 | 0.24 ** | 0.25 ** |
| ≥65 | 9.4 | 9.8 | 9.8 | 9.9 | 10.1 | 10.2 | 10.4 | 10.5 | 10.6 | 10.8 | 10.9 | 11.1 | 11.2 | 11.4 | 0.14 ** | 0.14 ** |
| **School years** | | | | | | | | | | | | | | | | |
| 0 to 8 | 45.5 | 45.0 | 43.7 | 42.0 | 40.6 | 38.8 | 36.8 | 36.6 | 35.9 | 34.6 | 32.5 | 30.8 | 30.2 | 28.8 | −1.32 ** | −1.38 * |
| 9 to 11 | 33.3 | 35.1 | 34.7 | 35.8 | 35.8 | 36.7 | 38.5 | 37.5 | 38.1 | 38.1 | 35.9 | 37.3 | 38.0 | 38.4 | 0.31 * | 0.35 |
| ≥12 | 21.2 | 19.8 | 21.6 | 22.2 | 23.5 | 24.5 | 24.7 | 25.9 | 25.9 | 27.3 | 31.6 | 31.9 | 31.8 | 32.8 | 1.01 ** | 1.04 |

Notes: Vigitel: Surveillance of Risk and Protective Factors for Chronic Diseases Telephone Suvey. * $p < 0.05$; ** $p < 0.001$; [†] Average annual variation. Corresponding to the Prais–Winsten regression coefficient of the variable over the survey year (expressed in pp per year). n = 730,309.

The prevalence of adults with hypertension was 24.5% on average in the period analyzed, and it remained stable from 2006 to 2019. According to age groups, there was a reduction in this prevalence among the youngest individuals (18–24 years old, with −0.08 pp/year), adults in the age groups 45–54 years (−0.21 pp/year), and 55–64 years (−0.25 pp/year). A significant increase in hypertension prevalence was observed for adults with up to 11 years of school (0–8 years: 0.65 pp/year; and 9–11 years: 0.35 pp/year). In the most recent period (2015–2019), this prevalence showed a significant reduction for the total population (−0.16 pp/year, it ranged from 25.1% to 24.6%), for men (−0.31 pp/year), for some age groups (the age group 35–44 years recorded the most significant reduction: −0.61 pp/year), and for adults with the highest schooling (12 years or more, with −0.18 pp/year). On the other hand, the increase in hypertension prevalence was observed among adults with the lowest schooling —from 0 to 8 years of school (0.48 pp/year) and from 9 to 11 years of school (0.16 pp/year) (Table 2).

In the adjusted analyses, adults with hypertension showed a significant association with unhealthy lifestyle habits: (i) lower recommended intake of fruits and vegetables (APR= 0.97; $p < 0.05$); (ii) lower regular intake of fruits (APR= 0.98; $p < 0.001$); (iii) lower regular intake of beans (APR = 0.97; $p < 0.001$); (iv) lower leisure-time exercising (APR = 0.89; $p < 0.001$); (v) higher abusive consumption of alcoholic beverages (APR = 1.04; $p < 0.05$); (vi) higher prevalence of overweight (APR = 1.40; $p < 0.001$); and (vii) higher prevalence of obesity (APR = 2.17; $p < 0.001$), confirmed by a higher mean BMI (APR = 1.10; $p < 0.001$) (Table 3). However, adults with hypertension also showed a significant association with three healthy lifestyle habits: (i) lower whole milk intake (APR = 0.93; $p < 0.001$); (ii) lower regular intake of sweets (APR = 0.85; $p < 0.001$); and (iii) lower smoking prevalence (APR = 0.83; $p < 0.001$) (Table 3).

**Table 2.** Prevalence [‡] of the adults (≥18 years old) with arterial hypertension in the capitals of the 26 Brazilian states and the Federal District), according to sociodemographic characteristics. Vigitel Brazil, 2006–2019.

| Variables | 2006 | 2007 | 2008 | 2009 | 2010 | 2011 | 2012 | 2013 | 2014 | 2015 | 2016 | 2017 | 2018 | 2019 | Coef. [†] (2006/19) | Coef. [†] (2015/19) |
|---|---|---|---|---|---|---|---|---|---|---|---|---|---|---|---|---|
| **Gender** | | | | | | | | | | | | | | | | |
| Male | 20.2 | 20.9 | 21.8 | 22.0 | 21.7 | 21.4 | 21.4 | 21.8 | 22.0 | 22.7 | 22.4 | 22.5 | 21.7 | 21.7 | 0.10 | −0.31 * |
| Female | 25.4 | 26.2 | 27.2 | 27.5 | 27.2 | 26.8 | 26.7 | 26.7 | 26.8 | 27.2 | 27.1 | 27.0 | 26.9 | 27.2 | 0.09 | −0.02 |
| **Age group, years** | | | | | | | | | | | | | | | | |
| 18 to 24 | 4.9 | 4.9 | 5.0 | 5.1 | 4.8 | 4.3 | 3.6 | 3.8 | 4.0 | 4.3 | 4.0 | 3.9 | 3.9 | 4.0 | −0.08 * | −0.07 |
| 25 to 34 | 9.7 | 9.9 | 10.7 | 10.3 | 9.9 | 9.0 | 8.7 | 8.8 | 9.2 | 9.7 | 9.9 | 9.2 | 9.2 | 8.8 | −0.08 | −0.28 * |
| 35 to 44 | 18.5 | 19.4 | 20.3 | 20.1 | 19.6 | 19.1 | 19.1 | 19.0 | 18.9 | 19.2 | 18.4 | 17.8 | 17.1 | 17.0 | −0.15 | −0.61 * |
| 45 to 54 | 33.6 | 34.9 | 35.6 | 35.7 | 34.7 | 34.8 | 34.4 | 33.8 | 33.6 | 33.5 | 32.7 | 32.3 | 31.5 | 32.2 | −0.21 * | −0.52 |
| 55 to 64 | 49.5 | 50.4 | 50.7 | 51.5 | 50.8 | 50.6 | 50.1 | 50.2 | 49.2 | 48.7 | 47.5 | 48.3 | 47.1 | 47.4 | −0.25 * | −0.29 * |
| ≥65 | 57.5 | 58.9 | 60.8 | 61.7 | 61.0 | 59.6 | 59.8 | 59.8 | 59.9 | 61.2 | 61.6 | 62.0 | 60.4 | 60.1 | 0.16 | −0.36 |
| **School years** | | | | | | | | | | | | | | | | |
| 0 to 8 | 32.8 | 34.2 | 36.3 | 37.3 | 37.1 | 36.9 | 37.4 | 38.0 | 38.7 | 39.9 | 40.5 | 41.3 | 41.2 | 42.0 | 0.65 ** | 0.48 * |
| 9 to 11 | 15.5 | 16.1 | 16.7 | 17.2 | 17.3 | 17.6 | 17.5 | 18.1 | 18.2 | 19.4 | 19.5 | 19.9 | 19.8 | 20.0 | 0.35 ** | 0.16 * |
| ≥12 | 14.0 | 14.2 | 14.5 | 14.4 | 14.5 | 14.5 | 14.7 | 14.5 | 14.8 | 14.9 | 15.0 | 14.7 | 14.5 | 14.3 | 0.03 | −0.18 * |
| **Total** | 23.0 | 23.8 | 24.7 | 25.0 | 24.7 | 24.3 | 24.2 | 24.4 | 24.6 | 25.1 | 24.9 | 24.9 | 24.5 | 24.6 | 0.09 | −0.16 * |

Notes: Vigitel: Surveillance of Risk and Protective Factors for Chronic Diseases Telephone Survey. * $p < 0.05$; ** $p < 0.001$; n = 730,309. [†] Average annual variation. Corresponding to the Prais–Winsten regression coefficient of the variable over the survey year (expressed in pp per year). [‡] Standardized by age and education and smoothed by moving average. For further information, see the methods section).

**Table 3.** Prevalence (% and 95%CI) and gross and adjusted prevalence ratio of health protective and risk factors, according to medical diagnosis of arterial hypertension. Vigitel Brazil, 2006–2019.

| Variables | Without Hypertension | | | | With Hypertension | | | | GPR | APR |
|---|---|---|---|---|---|---|---|---|---|---|
| | % | CI (95%) | | | % | CI (95%) | | | | |
| **Protective Factors** [§] | | | | | | | | | | |
| Recommended intake of fruits and vegetables | 22.3 | 22.1 | - | 22.6 | 23.7 | 23.3 | - | 24.1 | 1.06 ** | 0.97 * |
| Regular intake of fruits and vegetables | 33.4 | 33.1 | - | 33.7 | 37.7 | 37.2 | - | 38.1 | 1.13 ** | 0.99 |
| Regular intake of fruits | 57.5 | 57.2 | - | 57.8 | 62.5 | 62.0 | - | 62.9 | 1.09 ** | 0.98 ** |
| Regular intake of vegetables | 48.5 | 48.2 | - | 48.8 | 51.1 | 50.7 | - | 51.6 | 1.05 ** | 1.00 |
| Regular intake of beans | 65.1 | 64.8 | - | 65.4 | 63.3 | 62.8 | - | 63.7 | 0.97 ** | 0.97 ** |
| Leisure-time exercising | 37.9 | 37.6 | - | 38.2 | 26.2 | 25.8 | - | 26.6 | 0.69 ** | 0.89 ** |
| **Risk factors** [§] | | | | | | | | | | |
| Fat-rich meat intake | 33.4 | 33.0 | - | 33.7 | 26.5 | 26.0 | - | 27.0 | 0.79 ** | 0.98 |
| Whole milk intake | 56.4 | 56.1 | - | 56.8 | 49.1 | 48.6 | - | 49.7 | 0.87 ** | 0.93 ** |
| Regular intake of soft drinks and sugar-sweetened beverages | 23.5 | 23.2 | - | 23.7 | 17.1 | 16.7 | - | 17.5 | 0.73 ** | 1.01 |
| Regular intake of sweets | 20.7 | 20.3 | - | 21.1 | 13.4 | 12.8 | - | 13.9 | 0.65 ** | 0.85 ** |
| Abusive consumption of alcoholic beverages | 19.1 | 18.8 | - | 19.3 | 13.0 | 12.7 | - | 13.3 | 0.68 ** | 1.04 * |
| Smoking | 12.5 | 12.3 | - | 12.7 | 11.2 | 10.9 | - | 11.5 | 0.90 ** | 0.83 ** |
| Overweight | 44.4 | 44.2 | - | 44.7 | 68.2 | 67.8 | - | 68.6 | 1.53 ** | 1.40 ** |
| Obesity | 12.6 | 12.4 | - | 12.8 | 29.8 | 29.3 | - | 30.2 | 2.36 ** | 2.17 ** |
| BMI (mean) [¥] | 25.04 | 25.01 | - | 25.06 | 27.94 | 27.89 | - | 28.00 | 1.12 ** | 1.10 ** |

Notes: Vigitel: Surveillance of Risk and Protective Factors for Chronic Diseases Telephone Survey. GPR: Gross Prevalence ratio; APR: Adjusted prevalence ratio (sex, age, schooling level). n = 730,309. * $p < 0.05$; ** $p < 0.001$; [§] See Supplementary Materials. [¥] Mean of Body Mass Index (BMI).

Gender-stratified analysis validated the overall results recorded for the entire population. Except for recommended and regular intake of fruits and vegetables (APR = 0.97; $p < 0.05$ and APR = 0.97; $p < 0.001$, respectively), and for regular intake of vegetables (APR = 0.98; $p < 0.05$), which showed significant differences among women, and for regular intake of beans (APR = 0.95; $p < 0.001$) and abusive consumption of alcoholic beverages (APR = 1.1; $p < 0.001$), which showed significant difference among men. In addition, there

was a higher frequency of protective behaviors and a lower frequency of risk behaviors among women (with and without hypertension) than among men (Table 4).

**Table 4.** Prevalence (% and 95%CI) and crude and adjusted prevalence ratio of health-protective and risk factors, according to medical diagnosis of arterial hypertension, by sex. Vigitel Brazil, 2006–2019.

| Variables | Sex | Without Hypertension | | | | With Hypertension | | | | GPR | APR |
|---|---|---|---|---|---|---|---|---|---|---|---|
| | | % | CI (95%) | | | % | CI (95%) | | | | |
| **Protective factors** § | | | | | | | | | | | |
| Recommended intake of fruits and vegetables | Male | 18.0 | 17.6 | - | 18.3 | 18.6 | 18.0 | - | 19.3 | 1.04 | 0.97 |
| | Female | 26.3 | 26.0 | - | 26.7 | 27.2 | 26.7 | - | 27.7 | 1.03 * | 0.97 * |
| Regular intake of fruits and vegetables | Male | 27.2 | 26.8 | - | 27.6 | 30.5 | 29.7 | - | 31.2 | 1.12 ** | 1.00 |
| | Female | 39.0 | 38.7 | - | 39.4 | 42.6 | 42.0 | - | 43.2 | 1.09 ** | 0.97 ** |
| Regular intake of fruits | Male | 51.7 | 51.2 | - | 52.1 | 54.5 | 53.8 | - | 55.3 | 1.06 ** | 0.98 * |
| | Female | 62.8 | 62.4 | - | 63.1 | 68.0 | 67.4 | - | 68.5 | 1.08 ** | 0.98 ** |
| Regular intake of vegetables | Male | 42.8 | 42.4 | - | 43.3 | 45.3 | 44.5 | - | 46.1 | 1.06 ** | 1.01 |
| | Female | 53.7 | 53.4 | - | 54.1 | 55.2 | 54.6 | - | 55.7 | 1.03 ** | 0.98 * |
| Regular intake of beans | Male | 72.0 | 71.6 | - | 72.4 | 68.7 | 67.9 | - | 69.4 | 0.95 ** | 0.95 ** |
| | Female | 58.8 | 58.4 | - | 59.2 | 59.5 | 58.9 | - | 60.1 | 1.01 * | 0.99 |
| Leisure-time exercising | Male | 46.1 | 45.6 | - | 46.6 | 32.1 | 31.3 | - | 32.9 | 0.7 ** | 0.92 ** |
| | Female | 30.4 | 30.1 | - | 30.8 | 22.1 | 21.6 | - | 22.6 | 0.73 ** | 0.86 ** |
| **Risk factors** § | | | | | | | | | | | |
| Fat-rich meat intake | Male | 43.8 | 43.3 | - | 44.3 | 37.4 | 36.5 | - | 38.4 | 0.85 ** | 0.97 |
| | Female | 23.8 | 23.5 | - | 24.2 | 18.9 | 18.4 | - | 19.4 | 0.79 ** | 1.01 |
| Whole milk intake | Male | 59.1 | 58.6 | - | 59.6 | 50.9 | 50 | - | 51.9 | 0.86 ** | 0.92 ** |
| | Female | 54.0 | 53.5 | - | 54.4 | 47.9 | 47.2 | - | 48.6 | 0.89 ** | 0.93 ** |
| Regular intake of soft drinks and sugar-sweetened beverages | Male | 26.8 | 26.4 | - | 27.2 | 20.6 | 19.9 | - | 21.3 | 0.77 ** | 1.02 |
| | Female | 20.4 | 20.1 | - | 20.8 | 14.6 | 14.2 | - | 15.0 | 0.71 ** | 1.01 |
| Regular intake of sweets | Male | 17.8 | 17.2 | - | 18.4 | 12.9 | 12.0 | - | 13.8 | 0.72 ** | 0.89 * |
| | Female | 23.3 | 22.8 | - | 23.9 | 13.7 | 13.0 | - | 14.4 | 0.60 ** | 0.83 ** |
| Abusive consumption of alcoholic beverages | Male | 27.1 | 26.7 | - | 27.5 | 22.4 | 21.7 | - | 23.1 | 0.83 ** | 1.10 ** |
| | Female | 11.7 | 11.5 | - | 12.0 | 6.5 | 6.3 | - | 6.8 | 0.56 ** | 0.95 |
| Smoking | Male | 15.5 | 15.2 | - | 15.9 | 13.8 | 13.2 | - | 14.4 | 0.89 ** | 0.83 ** |
| | Female | 9.7 | 9.4 | - | 9.9 | 9.4 | 9.1 | - | 9.8 | 0.98 | 0.82 ** |
| Overweight | Male | 49.4 | 49.0 | - | 49.9 | 70.9 | 70.2 | - | 71.7 | 1.44 ** | 1.35 ** |
| | Female | 39.9 | 39.5 | - | 40.3 | 66.3 | 65.8 | - | 66.8 | 1.66 ** | 1.42 ** |
| Obesity | Male | 12.9 | 12.6 | - | 13.2 | 29.0 | 28.3 | - | 29.7 | 2.25 ** | 2.21 ** |
| | Female | 12.4 | 12.2 | - | 12.7 | 30.3 | 29.8 | - | 30.8 | 2.44 ** | 2.08 ** |
| BMI (mean) ¥ | Male | 25.45 | 25.41 | - | 25.49 | 27.96 | 27.88 | - | 28.05 | 1.10 ** | 1.09 ** |
| | Female | 24.66 | 24.62 | - | 24.69 | 27.93 | 27.85 | - | 28.01 | 1.13 ** | 1.10 ** |

Notes: Vigitel: Surveillance of Risk and Protective Factors for Chronic Diseases Telephone Survey. GPR: Gross Prevalence ratio; APR: Adjusted prevalence ratio (age, schooling level). * $p < 0.05$; ** $p < 0.001$; n = 730.309. § See Supplementary Materials. ¥ Mean of Body Mass Index (BMI).

## 4. Discussion

Based on data about more than 700,000 individuals between 2006 and 2019, we analyzed the temporal evolution of hypertension prevalence and compared the prevalence of health behaviors among Brazilian adult individuals with and without hypertension. Hypertension prevalence among Brazilian adults remained stable over the entire investigated period (2006–2019). However, the most recent period (2015–2019) recorded a significant decrease in such a prevalence. Both periods recorded reduced hypertension prevalence among the youngest individuals and among those in the age group 55–64 years, as well as increased prevalence of it among those with the lowest schooling. Individuals with hypertension presented a lower prevalence of protective factors—such as recommended intake of fruits and vegetables, regular intake of fruits, regular intake of beans, and leisure-time exercising—as well as a higher prevalence of health-risk behaviors—such as abusive consumption of alcoholic beverages, overweight, and obesity—than adult individuals without hypertension.

The trend of hypertension prevalence stability among adult individuals in the total investigated period, as well as of reduced hypertension prevalence in the most recent period (2015 to 2019), was like that observed in high-income countries [4,5]. From the 1980s onwards, economically developed countries have shown hypertension prevalence stability, although there was a decrease in such a prevalence in some age groups. The justification often observed for such a decrease lies in increased awareness of and improved living and health conditions, mainly among older individuals. Increased awareness of the population has stimulated the search for medical care, resulting in an increase in the number of diagnoses at initial moments, reflecting an increase in the prevalence of hypertension, but resulting in a reduction of this prevalence at a later [4]. Nationally implemented health screening and check-up programs (disease prevention examinations) and stricter control targets may also have contributed to the observed improvements [4].

The presence of a higher level of education can favor a better quality of life and greater access to health information, which could be associated with the adoption of healthier lifestyles [15]. Considering that risk behaviors can lead to different NCDs, non-adherence to lifestyle changes and medical treatment for hypertension and other NCDs worsens the prognosis of these diseases.

It is noteworthy that although Brazil is a middle-income country, hypertension prevalence in it is not similar to that in countries with the same economic features that have shown increased hypertension prevalence in most cases [5,6]. The better evolution observed for trends associated with hypertension prevalence in Brazil, mainly in the most recent period, may be explained by the availability of a universal and free-of-charge public health system in the country (the Brazilian Unified Health System (SUS), implemented in 1989), as well as by improvements made in health monitoring and care policies from the early 2000s onwards [16,17].

The development of the primary health care (PHC) system in the country was a determining factor capable of improving the Brazilian population's health condition, with emphasis on programs aimed at healthcare municipalization and expansion processes, such as the Family Health Program [17]. In 2001, the MoH implemented the "Plan for the Reorganization of the Care of Arterial Hypertension and Diabetes Mellitus", which aimed at establishing the guidelines and goals for the reorganization of hypertension care within SUS in order to reduce morbidity and mortality rates resulting from CVD and to improve the quality of life of the Brazilian population [16,17]. Among the Plan's propositions are better training for health professionals, promoting educational activities, improving pharmaceutical care, continuously monitoring ill individuals, and screening new cases of hypertension [18].

In 2011, the elaboration of the "Strategic Action Plan to Tackle NCD in Brazil 2011–2022" defined new goals to mitigate hypertension-related morbidity and mortality rates; among them, expanding pharmaceutical care, developing educational material to train primary healthcare professionals on how to treat hypertensive patients, and adopting intense health promotion strategies to reduce risk factors for NCDs [10]. The Plan aims to reduce premature death resulting from CVD by 2% a year by 2022 [10]. In 2014, the document titled "Strategies for the care of the person with chronic disease: systemic arterial hypertension" was published by the MoH to improve prevention actions and to help in managing the actions of PHC professionals [19]. These professionals work in teams accountable for identifying hypertension risk factors, as well as for implementing healthcare promotion and disease prevention actions to help improve patients' quality of life and to encourage their adherence to proper treatments [20].

The proper treatment for individuals with hypertension includes not only drug therapy (executed when necessary) but mainly the adoption of healthy behaviors (that must be reinforced regardless of the disease diagnosis) [21]. However, unhealthy behaviors can be observed even after hypertension diagnosis, a fact that contributes to a lack of control over blood pressure levels or even to the emergence of more severe CVDs [22]. Blood pressure control enables a better quality of life for individuals with hypertension, being

essential for their management. Nonetheless, the data are alarming worldwide, in which about 1 in 5 adults (21%) with hypertension has it under control [23]. This scenario can still be aggravated by the environmental context in which the individual finds himself. The characteristics of the urban physical environment—pollution, mass public transport, high population density, and reduced green area—contribute to several factors that may be related to NCDs and hypertension [24]. We highlight this fact in relation to our sample of adults from Brazilian capitals. The urbanized environment can influence the sample.

Based on the current findings, it was possible observing a higher frequency of alcohol abuse and insufficient leisure-time exercising among adult individuals with hypertension, in association with a high prevalence of overweight and obesity. Behavioral patterns that mostly comprise risk factors tend to result in a worse prognosis for ill individuals and a higher risk of developing hypertension among healthy individuals. Changes in lifestyle behaviors, such as weight loss, healthier food intake (diets rich in fruits, vegetables, and whole grains), avoiding alcohol intake, exercising on a regular basis, and smoking cessation, can result in better blood pressure control [21].

Some limitations must be taken into consideration. The first one refers to the use of telephone interviews only conducted in Brazilian state capitals. A series of studies comparing indicators deriving from Vigitel to indicators deriving from household surveys have indicated the methodological robustness of the system, as well as its ability to predict the behavior of indicators measured for the whole population living in each investigated city [25–27]. It happens due to the methodology used by the system, which is based on official residential telephone records, as well as the application of weighting factors based on official population projections [12]. Another limitation of it lies in the use of self-reported information, which could reduce the accuracy of results. However, information based on the condition reported by respondents has been widely used in large health surveys due to its promptness, low cost, and satisfactory validity [28] to investigate and monitor risk factors among different countries [29]. Finally, the short period investigated represents a relatively limited historical series (fourteen years). Temporal variations that do not show uniformity or that show small magnitude tend not to be identified by the analytical procedures used in the current study.

## 5. Conclusions

Hypertension prevalence in the Brazilian population remained stable over the entire investigated period, although it decreased in the most recent period. Individuals with hypertension reported unfavorable scenarios for healthy habits when compared to adult individuals without hypertension. These results reinforce the need to expand and strengthen public health actions focused on individuals with hypertension to encourage the population to adopt healthy lifestyles.

**Supplementary Materials:** The following supporting information can be downloaded at: https: //www.mdpi.com/article/10.3390/obesities3020012/s1, Table S1: Health protective and risk factors analyzed and period for which the indicator is available. Vigitel Brazil, 2006–2019.

**Author Contributions:** Conceptualization, T.C.M.C. and R.M.C.; Methodology, T.C.M.C., A.C.R.A.S., and R.M.C.; Formal Analysis, T.C.M.C. and R.M.C.; Writing—Original Draft Preparation, T.C.M.C., A.C.R.A.S., M.M.S., and E.G.M.; Writing—Review and Editing, all authors. All authors have read and agreed to the published version of the manuscript.

**Funding:** The work was supported by the Coordenação de Aperfeiçoamento de Pessoal de Nível Superior (CAPES) (grant number 001 (Scholarship for M.M.S.)) and the Conselho Nacional de Desenvolvimento Científico e Tecnológico (grant number 311170/2019-6 (Scholarship for R.M.C.)).

**Institutional Review Board Statement:** Vigitel's was authorized by the Brazilian Committee on Ethics in Research with Human Beings of the Ministry of Health (65610017.1.0000.0008).

**Informed Consent Statement:** Oral informed consent was obtained from all subjects involved in the study during the interview.

**Data Availability Statement:** Vigitel's data are available on the official Ministry of Health website: http://svs.aids.gov.br/download/Vigitel/ (accessed on 3 February 2023).

**Conflicts of Interest:** The authors declare no conflict of interest.

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
