# Peer review of "Trend in Hypertension Prevalence and Health Behaviors among the Brazilian Adult Population: 2006–2019"

_2673-4168, doi:10.3390/obesities3020012_

Round 1
Reviewer 1 Report
The article investigated the prevalence of hypertension over a 14 year period in Brazil. The evidence is well recorded and the authors provide sufficient information. One question is, did they collect information about income. Do those with higher education have higher income and how does this affect their ability to afford more healthy food. Does their education inform their lifestyle choices? Please review the paper for some minor corrections.
line 12-13: Rephrase the first sentence.
line 31: Statistical reference is 2016. Is there more recent data available?
line 42: Rephrase sentence.
line 100: "(=150 minutes/week (moderate intensity) or 75 minutes/week (vigorous intensity))" Is this correct?
line 219: Rephrase sentence.
line 259 and 265: remove "one finds"
line 279: Rephrase sentence.
line 281: Rephrase sentence.
line 286 and 308: Full-stop missing.
Author Response
Response to Reviewer 1 Comments
We are grateful for the evaluation carried out by the Obesities in the present manuscript. We submitted a revised version of it, in agreement with the suggestions from the reviewer 1, as described below. The changes made in the manuscript.
We are available to provide any further information, if needed.
Kind Regards,
Corresponding author.
Reviewer 1: The article investigated the prevalence of hypertension over a 14 year period in Brazil. The evidence is well recorded and the authors provide sufficient information. One question is, did they collect information about income. Do those with higher education have higher income and how does this affect their ability to afford more healthy food. Does their education inform their lifestyle choices? Please review the paper for some minor corrections.
Authors: Unfortunately, information on income was not collected in the Vigitel survey, making it impossible for us to use this information in our study. The relationship between higher education and better health habits has already been reported in other studies. We have added the text below in the manuscript.
“The presence of a higher level of education can favor a better quality of life and greater access to health information, which could be associated with the adoption of healthier lifestyles [16]. Considering that risk behaviors can lead to different NCDs, non-adherence to lifestyle changes and medical treatment for hypertension and other NCDs worsens the prognosis of these diseases.”
Reviewer 1: line 12-13: Rephrase the first sentence.
Authors: We rewrote the objective.
“Our objective was to analyze temporal trends in the prevalence of self-reported hypertension among Brazilian adults and to investigate differences in health behaviors between individuals with and without hypertension between 2006 and 2019.”
Reviewer 1: line 31: Statistical reference is 2016. Is there more recent data available?
Authors: We revised the sentence in the introduction and replaced it with a more recent one.
“Hypertension is the most prevalent preventable risk factor for cardiovascular diseases (CVD), which are the leading causes of death in the Region of the Americas with 2.0 million deaths from CVC in 2019”.
Reviewer 1: line 42: Rephrase sentence.
Authors: We rewrote the sentence.
“Controlling and reducing the prevalence of hypertension by 25% in the population between 2015 and 2025, is one of the goals of the "Global Action Plan for the Prevention and Control of NCDs 2013-2020" by the World Health Organization (WHO) [9].”
Reviewer 1: line 100: "(=150 minutes/week (moderate intensity) or 75 minutes/week (vigorous intensity))" Is this correct?
Authors: The indicator is correct. According to the WHO, an adult individual (≥ 18 years old) needs to perform at least 150 minutes of PA per week with a moderate intensity or 75 minutes per week with a vigorous intensity (with minimum uninterrupted time equal to 10 minutes) for provide health benefits and effective prevention against NCDs. Based on this recommendation, since 2011, the indicator “sufficient practice of LTPA (≥150 minutes/ week) (Yes | No)” has been used by Vigitel, presenting good reproducibility and sufficiently accurate. WHO reference was inserted to support the indicator.
Reviewer 1: line 219: Rephrase sentence.
Authors: We rewrote the sentence.
“Based on data about more than 700,000 individuals between 2006 and 2019, we analyzed the temporal evolution of hypertension prevalence and comparing the prevalence of health behaviors among Brazilian adult individuals with and without hypertension.”
Reviewer 1: line 259 and 265: remove "one finds"
Authors: We have removed these words.
Reviewer 1: line 279: Rephrase sentence.
Authors: We rewrote the sentence.
“The blood pressure control enables a better quality of life for the individuals with hypertension, being essential for their management.”
Reviewer 1: line 281: Rephrase sentence.
Authors: We rewrote the sentence.
“Nonetheless the data are alarming worldwide, in which, about 1 in 5 adults (21%) with hypertension has it under control [23].”
Reviewer 1: line 286 and 308: Full-stop missing.
Authors: We insert the full-stop.
Reviewer 2 Report
Trend in hypertension prevalence and health behaviors among 2 the Brazilian adult population: 2006 – 2019, is a good title and a large analyze in your country.
In abstract I cannot see the object of your study, and especially what is new in the field. Add the conclusion, also. In the results, I would like to see the survival rate, or the classification by cardiac damage, and correlations with age groups or diet. table 3 should also contain BMI and degree of obesity, for comparative fatness. We need a more complex expertise to be able to bring news, considering that your batch is large, and the time duration is sufficient for the survival rate. in this context, the discussions and conclusions must be completed. I wish you success and congratulate you on your activity.
Author Response
Response to Reviewer 2 Comments
We are grateful for the evaluation carried out by the Obesities in the present manuscript. We submitted a revised version of it, in agreement with the suggestions from the reviewer 2, as described below. The changes made in the manuscript.
We are available to provide any further information, if needed.
Kind Regards,
Corresponding author.
Trend in hypertension prevalence and health behaviors among 2 the Brazilian adult population: 2006 – 2019, is a good title and a large analyze in your country.
Reviewer 2: In abstract I cannot see the object of your study, and especially what is new in the field.
Authors: We rewrote the objective. The study is a temporal analysis of the prevalence of hypertension in Brazil with more than 700 thousand adults interviewed for more than 14 years. The analysis of these data allows us to present the Brazilian scenario regarding health actions for the control of Hypertension.
“Our objective was to analyze temporal trends in the prevalence of self-reported hypertension among Brazilian adults and to investigate differences in health behaviors between individuals with and without hypertension between 2006 and 2019.”
Reviewer 2: Add the conclusion, also.
Authors: We add the conclusion in the abstract.
“Hypertension prevalence has remained stable in entire period and decreased in the most recent period. Individuals with hypertension reported unfavorable scenario for healthy habits.”
Reviewer 2: In the results, I would like to see the survival rate, or the classification by cardiac damage, and correlations with age groups or diet.
Authors: Although this is a temporal series analysis, our study is based on cross-sectional data from the Brazilian adult population. It is not possible to carry out additional analyzes to calculate the survival rate.
Reviewer 2: table 3 should also contain BMI and degree of obesity, for comparative fatness.
Authors: We added the mean Body Mass Index (BMI) in Table 3 and Table 4.
Reviewer 2: We need a more complex expertise to be able to bring news, considering that your batch is large, and the time duration is sufficient for the survival rate. in this context, the discussions and conclusions must be completed.
Authors: Our study is based on cross-sectional data from the Brazilian adult population. Therefore, it is not possible to carry out additional analyzes to calculate the survival rate. We changed the study nomenclature to temporal trends analysis to reduce confusion.
Round 2
Reviewer 2 Report
Table 3 and 4
please explain the term Overweight and Obesity
Conclusions of your study don`t bring anything new, you can improve it specifically.